# Within-Week Variations and Relationships between Internal and External Intensities Occurring in Male Professional Volleyball Training Sessions

**DOI:** 10.3390/ijerph19148691

**Published:** 2022-07-17

**Authors:** Ricardo Franco Lima, Francisco Tomás González Férnandez, Ana Filipa Silva, Lorenzo Laporta, Henrique de Oliveira Castro, Sérgio Matos, Georgian Badicu, Gonçalo Arezes Pereira, Gustavo De Conti Teixeira Costa, Filipe Manuel Clemente

**Affiliations:** 1The Research Centre of Sports Performance, Recreation, Innovation and Technology (SPRINT), 4960-320 Melgaço, Portugal; anafilsilva@gmail.com (A.F.S.); sfilipematos@esdl.ipvc.pt (S.M.); filipe.clemente5@gmail.com (F.M.C.); 2Escola Superior Desporto e Lazer, Instituto Politécnico de Viana do Castelo, 4900-347 Viana do Castelo, Portugal; goncalomanuelarezes@hotmail.com; 3Department of Physical Education and Sport, Faculty of Education and Sport Sciences, Campus Melilla, University of Granada, 52006 Melilla, Spain; ftgonzalez@ugr.es; 4Sports Sciences, Health Sciences and Human Development (CIDESD), 5001-801 Vila Real, Portugal; 5Centro de Educação Física e Desportos, Univerisidade Federal de Santa Maria, Santa Maria 97105-900, Brazil; laporta.lorenzo@ufsm.br; 6Faculdade de Educação Física, Universidade Federal de Mato Grosso, Cuiabá 78068-600, Brazil; henriquecastro88@yahoo.com.br; 7Department of Physical Education and Special Motility, Faculty of Physical Education and Mountain Sports, Transilvania University of Brasov, 500068 Brasov, Romania; georgian.badicu@unitbv.ro; 8Faculdade de Educação Física e Dança, Universidade Federal de Goiás, Goiânia 74690-900, Brazil; conti02@ufg.br; 9Instituto de Telecomunicações, Delegação da Covilhã, 1049-001 Lisbon, Portugal

**Keywords:** volleyball workload, internal load, external load, monitoring, intensity, elite male

## Abstract

The purpose of the study was to test the within-week variations of the internal and external training intensity outcomes organized by days of the week. An 8-month observational period was conducted during the 2020–2021 season. The training sessions and matches of an elite volleyball team were monitored daily. The data comes from 14 players (two setters, five middle blockers, five outside hitters, and two opposites) of an elite team from the Portuguese 1st League (age: 21.7 ± 4.19 years of age; experience: 6.2 ± 3.8 years; body mass: 85.7 ± 8.69 kg; height: 192.4 ± 6.25 cm; BMI: 23.1 ± 1.40 kg/m^2^). The CR10 Borg scale was applied daily to measure the training intensity. The rate of perceived exertion (RPE) and the session-RPE were extracted as the internal outcomes. The external intensity was measured using an inertial measurement unit (IMU). The number of jumps, height average of jumps (JHA), minimum jump (MJ), maximal jump (MXJ), range jump (RJ), number of jumps (NJ), and training session density (D) were extracted as external intensity outcomes. The results showed that there was a difference between RPE and S-RPE (F (1.98) = 6.31, *p* = 0.01, η^2^ = 0.36, and F (1.73) = 28.30, *p* = 0.001, η^2^ = 0.72), as well as JHA and NJ (F (2.14) = 4.76, *p* = 0.02, η^2^ = 0.30, and F (1.77) = 4.77, *p* = 0.02, η^2^ = 0.30) within the microcycle. When analyzing the correlations between internal and external intensity, it was observed that there was a negative correlation between the Maximum Jump (4, 3, and 1 days before the Match day) (r^2^ = 0.34, r^2^ = 0.40, r^2^ = 0.41, respectively) and the Range Jump (3 and 1 days before the Match day (r^2^ = 0.33, r^2^ = 0.38, respectively) with the RPE (4 days before the Match day) and Maximum Jump (5, 4, 3, and 1 days before the Match day (r^2^ = 0.35, r^2^ = 0.39, r^2^ = 0.44, r^2^ = 0.34, respectively) and Range Jump (5, 4, 3, and 1 days before the Match day) (r^2^ = 0.34, r^2^ = 0.35, r^2^ = 0.40 and r^2^ = 0.36, respectively) with S_RPE (4 days before the Match day). Such findings show that higher internal intensities are correlated with lower external intensities in sessions further away from the game day. Such results could be an important tool for coaches to reflect, plan, monitor, and execute the training unit according to the temporal distance to the competition.

## 1. Introduction

Monitoring the internal and external intensities [1] has been used to identify the quality and quantity of training demands in elite team sports [2,3,4], and is an important process for assessing fatigue, recovery, and physical adaptations and avoiding injury risk [5,6,7,8]. Internal intensity refers to psychological and physiological stress, monitored through heart rate (HR), the rating of perceived exertion (RPE), and session-RPE, among others [9,10]. External intensity represents a work estimate performed by a player [11] and refers to the total amount of locomotive or mechanical stress generated by an athlete during exercise, monitored through the total distance traveled and the number of jumps a volleyball player undertakes, among others [7,10]. In this way, the relationships between internal and external intensities can serve to assess the athlete’s physical fitness during a specific exercise [2].

In this sense, the session-RPE method [12] is presented as a simple, low-cost strategy for quantifying the internal training intensity in team sports, including volleyball [13,14]. Specifically, volleyball involves a combination of multidirectional movements, repetitive and different jumps, and long matches, which require control and training process adjustment [4,15,16,17]. Furthermore, this team sport presents intermittent characteristics, which require that the athletes perform short-duration and high-intensity efforts interspersed with low-intensity periods [14]. In recent years, studies on men’s volleyball have been carried out using the RPE and session-RPE to monitor the internal intensity and jump analysis to monitor the external intensity [4,15,16,18].

In the recent study carried out by Lima et al. [16], the internal (using the RPE and session-RPE) and external (jumps number and height) intensities of elite male volleyball athletes were evaluated, and it was demonstrated that the RPE and number of jumps are higher in sessions further away from matchday, decreasing as match day (MD) approaches. For internal intensity, Bara Filho et al. [14] compared and correlated different methods of internal intensity control in volleyball players, concluding the session-RPE method better reflects the training loads, making it more reliable to control the training monitoring. For the external intensity, Lima et al. [19] evaluated elite male athletes from different playing positions in relation to the number and height of jumps and demonstrates the setter made a significantly greater number of jumps (n ≅ 32) than middle blockers (n ≅ 21) and outside hitters (n ≅ 13); however, no difference was found in the jump intensities across playing positions.

Few studies have analyzed the relationship between external and internal intensities in elite volleyball players [16]. Volleyball training must result from the intensity demands, considering the players’ specificity and their tasks in the match [20], making it essential to monitor and control the relationships between internal and external intensities daily [2,16] to balance training and match demands and avoid overload and possible injuries in athletes [5,8,21]. Considering the relevance of understanding the influence of the day of the week on the magnitude of the training intensity and also understanding how the external intensities may constrain the physiological responses, the current study aimed to (i) test the within-week variations of the internal and external training intensity outcomes and (ii) test the relationships between internal and external training intensity outcomes organized by days of the week. We hypothesize that significant differences in external and internal training intensity within the week will be observed. Specifically, the training sessions in the middle of the week will be those with higher efforts reported. In the case of the second objective, we hypothesize that internal intensities reported subjectively (e.g., rate of perceived exertion) will present significant correlations with the number of jumps since they may be modulated by the metabolic stimulus.

## 2. Materials and Methods

### 2.1. Experimental Approach to the Problem

An observational and longitudinal study was adopted. Data were obtained from 8 months of training and matches from the 2020–2021 season in a volleyball team that contained 14 players within the squad. During the competitive period, a total of 101 intra-squad training sessions matches and 26 competitive matches were conducted. Furthermore, 89,801 observations were gathered from training matches and 11,214 observations were gathered from matches.

The data contain players’ position, training/match duration, and training/match external and internal workload. Internal load refers to RPE and session-RPE (s-RPE) and external load refers to Jump Heigh Average (JHA), minimum jump (MJ), maximal jump (MXJ), range jump (RJ), number of jumps (NJ), and training session density (D). Data also contain a variable called the microcycle, which indicates the distance to the next game with possible values of 5 days before the match day (MD-5), 4 days before the match day (MD-4), 3 days before the match day (MD-3), 2 days before the match day (MD-2), and 1 day before the match day (MD-1).

### 2.2. Participants

For this study, fourteen elite male volleyball athletes (two setters, five middle blockers, five outside hitters, and two opposites) of an elite team from the Portuguese 1st League (age: 21.7 ± 4.19 years of age; experience: 6.2 ± 3.8 years; body mass: 85.7 ± 8.69 kg; height: 192.4 ± 6.25 cm; BMI: 23.1 ± 1.40 kg/m^2^) were included. Athletes were monitored throughout the entire 2020/21 season of the Portuguese Championship, corresponding to a total of 26 microcycles, 101 training sessions, 23 championship matches (which had 4 congested games), and 3 Portuguese Cup matches. In each microcycle with a single match, athletes often practice 5 to 7 times a week, while in congested fixtures (2 matches on the weekend), they often practice 3 to 4 times a week. The following inclusion criteria were used throughout the competitive period: (i) Players did not have injuries or illnesses during the period of data collection.

All athletes voluntarily participated in the study and were informed about the study’s design, implications, risks, and benefits. After receiving study information, the players provided informed consent. The study was conducted in line with the international ethical guidelines for sport and exercise science research recommended by the Declaration of Helsinki (2013). In addition, the study protocol was approved by the ethics committee at Escola Superior de Desporto e Lazer do Instituto Politécnico de Viana do Castelo (CTC-ESDL-CE003-2020).

### 2.3. Instruments

#### 2.3.1. Internal Intensity

The CR10 Borg scale [22] was used 20 min after practice sessions in the evening, and athletes had two weeks of familiarization (applied daily across the pre-competitive phase) and were asked to answer the following question: “How hard was your workout?”. This is a subjective scale with a simple procedure to apply, which has been used to monitor the internal workload in sports [16,23,24,25,26,27].

Furthermore, we calculated, for each session, the rating of perceived exertion-based training load (s-RPE) as arbitrary units (A.U.) and considered it to be the product between the Borg scale with the training session time [18] (s-RPE = Borg Scale Value * Time of Training session). The s-RPE was chosen according to the validity and reliability tested and applied in several studies with intraclass correlations that ranged between 0.8 and 0.9 points [16,23,24,25,26,27]. These previous studies found that to monitor and assess the internal load with a simple and cheap method, the CR10 Borg scale and the S_RPE constitute a consistent tool to provide individual and collective data to coaches.

#### 2.3.2. External Intensity

The inertial measurement unit (IMU) accelerometer VERT device (Vert^®^Classic, MyVert, Fort Lauderdale, FL, USA) (mean bias of 3.57 cm and 4.28 cm) [28] has been used in recent studies [4,16,29] and was used here to provide the number and height of jump actions during the training session. Taking advantage of the data collected from the device, we also opted to measure the density of the training sessions [30]. The density of the training session is considered as the frequency of jumps divided by the period of time (number of jumps/time of training session).

### 2.4. Procedures

Data were collected throughout the competitive phase. To measure the external intensity, in every training session, after warming up, players wore a belt with the Vert device until the end of the practice. It is relevant to say that the data were collected during the main phase of the practice session, excluding all bias coming from the warm-up workload. The devices were controlled and transmitted via Bluetooth in real-time through the MyVert App (from IOS). At the end of the practice session, data were exported to a spreadsheet. Depending on the competitive period phase, the training sessions per week varied between 3 and 7 practices (105 ± 12.4 min). The evening practices always started at 7 P.M., while the morning training sessions always started at 10 A.M.

### 2.5. Statistical Analysis

For data processing, the mean and standard deviation were used. Descriptive statistics were calculated for each variable. The Shapiro–Wilk test was conducted to verify whether all data were normally distributed. RPE, s-RPE, JHA, MJ, MXJ, RJ, NJ, and D were analyzed using a repeated-measures ANOVA for MD-5, MD-4, MD-3, MD-2, and MD-1. Effect size is indicated with partial eta squared for Fs. Finally, multiple pairwise comparisons were employed to obtain the differences between conditions, and the Bonferroni correction was used to compensate for the multiple post hoc comparisons. Effect size is indicated with Cohen’s d for pairwise comparisons and partial eta squared for Fs. Posteriorly, a Pearson correlation coefficient r was used to examine the relationship between values of internal load and values of external load, and to interpret the magnitude of these correlations, we adopted the following criteria: r ≤ 0.1, trivial; 0.1 < r ≤ 0.3, small; 0.3 < r ≤ 0.5, moderate; 0.5 < r ≤ 0.7, large; 0.7 < r ≤ 0.9, very large; and r > 0.9, almost perfect [29]. In addition, the regression analysis was used to identify which values of internal load can better explain the values of external load. The magnitude of r^2^ was interpreted as follows: >0.02, small; >0.13, medium; >0.23, large. The inflation factors of the variance were calculated to verify that the collinearity was not a serious concern. The data were analyzed using the software Statistics (version 13.1; Statsoft, Inc., Tulsa, OK, USA) and the alpha level was set at *p* < 0.05.

## 3. Results

Descriptive statistics were calculated for each internal intensity (RPE and session-RPE) and external intensity variables (Jump Height Average, Minimum Jump, Maximal Jump, Range Jump, Number of Jumps, and Training Session Density) (see Table 1).

First, a different repeated-measures ANOVA with participants’ mean internal load (RPE and S-RPE) revealed a significant effect in both cases, F (1.98) = 6.31, *p* = 0.01, η^2^ = 0.36, and F (1.73) = 28.30, *p* = 0.001, η^2^ = 0.72, respectively. Second, a new repeated-measures ANOVA with participants’ mean external load revealed a significant effect for JHA and NJ, F (2.14) = 4.76, *p* = 0.02, η^2^ = 0.30, and F (1.77) = 4.77, *p* = 0.02, η^2^ = 0.30. However, the dataset was not significant for MJ, MXJ, RJ, and D, F (1.63) = 1.52, *p* = 0.24, η^2^ = 0.12, F < 1, F < 1, and F (2.40) = 2.58, *p* = 0.09, η^2^ = 0.19, respectively.

At this point, post hoc comparisons with internal intensity (RPE and S-RPE) showed significant differences between MD-5 and MD-4, *p* = 0.05, d = 0.78, MD-5 and MD-3, *p* = 0.001, d = 1.06, MD-5 and MD-2, *p* = 0.01, d = 1.12, MD-5 and MD-1, *p* = 0.001, d = 1.96, MD-4 and MD-1, *p* = 0.001, d = 1.33, d = 0.17, and MD-3 and MD-1, *p* = 0.001, d =1.51. However, in the case of MD-3 vs. MD-2, MD-4 vs. MD-3, and MD-4 vs. MD-2, it did not reveal significant differences.

In this sense, another post hoc comparison with external intensity (MJ, RJ, and D) did not reveal significant differences between any MD. Nevertheless, post hoc comparisons with JHA revealed significant differences between MD-5 and MD-3, *p* = 0.05, d = −0.05, MD-5 and MD-2, *p* = 0.03, d = 0.06, MD-5 and MD-1, *p* = 0.001, d = −0.07, MD-4 and MD-3, *p* = 0.03, d = −0.02, d = 0.17, and MD-4 and MD-1, *p* = 0.03, d = −0.03. However, the comparison between MD-5 and MD-4, MD-4 and MD-2, MD-3 and MD-2, MD-3 and MD-2, and MD-2 and MD-1 revealed no differences. In the same direction, another post hoc comparison with MXJ revealed significant differences between MD-5 and MD-4, *p* = 0.001, d = 0.02, MD-5 and MD-3, *p* = 0.001, d = 0.02, MD-5 and MD-2, *p* = 0.001, d = 0.02, MD-5 and MD-1, *p* = 0.001, d = 0.03, MD-4 and MD-2, *p* = 0.02, d = 0.01, MD-4 and MD-1, *p* = 0.001, d = 0.01, MD-3 and MD-2, *p* = 0.05, d = 0.01, MD-3 and MD-1, *p* = 0.001, d = 0.03, and MD-2 and MD-1, *p* = 0.001, d = 0.02. However, the comparison between MD-4 and MD-3 did not reveal significant differences. Last, a new post hoc comparison with NJ showed significant differences between MD-5 and MD-1, *p* = 0.001, d = 0.03, MD-4 and MD-2, *p* = 0.03, d = 0.01, MD-4 and MD-1, *p* = 0.001, d = 0.03, MD-3 and MD-1, *p* = 0.001, d = 0.03, and MD-2 and MD-1, *p* = 0.001, d = 0.02. Notwithstanding, the comparison between MD-5 and MD-4, MD-5 and MD-3, MD-5 and MD-2, MD-4 and MD-3, and MD-3 and MD-2 did not reveal significant differences (see Table 1 for more information).

At this point, a correlation analysis was performed between participants’ mean internal intensity (RPE and S-RPE) and participants’ mean external intensity (JHA, MJ, MXJ, RJ, NJ, and D) for MD-5, MD-4, MD-3, MD-2, and MD-1.

### 3.1. Jump Height Average

Correlation analyses were performed between JHA (JHA-5, JHA-4, JHA-3, JHA-2, and JHA-1) and RPE (RPE-5, RPE-4, RPE-3, RPE-2, and RPE-1) and S-RPE (S-RPE-5, S-RPE-4, S-RPE-3, S-RPE-2, and S-RPE-1), which did not show any correlations.

### 3.2. Minimum Jump

Another correlation analysis was performed between MJ (MJ-5, MJ-4, MJ-3, MJ-2, and MJ-1) and RPE (RPE-5, RPE-4, RPE-3, RPE-2, and RPE-1) and S-RPE (S-RPE-5, S-RPE-4, S-RPE-3, S-RPE-2, and S-RPE-1), which did not reveal any correlations.

### 3.3. Maximal Jump

A new correlation analysis was performed between MXJ (MXJ-5, MXJ-4, MXJ-3, MXJ-2, and MXJ-1) and RPE (RPE-5, RPE-4, RPE-3, RPE-2, and RPE-1) and S-RPE (S-RPE-5, S-RPE-4, S-RPE-3, S-RPE-2, and S-RPE-1). Data showed large negative correlations between MXJ-4 and RPE-4 (r = −0.59. *p* = 0.04), MXJ-3 and RPE-4 (r = −0.64. *p* = 0.02), and MXJ-1 and RPE-4 (r = −0.65. *p* = 0.02). In this sense, another large negative correlation was found between MXJ-5 and S-RPE-4 (r = −0.59. *p* = 0.04), MXJ-4 and S-RPE-4 (r = −0.63. *p*= 0.03), MXJ-3 and S-RPE-4 (r = −0.67. *p* = 0.01), and MXJ-1 and S-RPE-4 (r = −0.59. *p* = 0.04).

### 3.4. Range Jump

Another correlation analysis was performed between RJ (RJ-5, RJ-4, RJ-3, RJ-2, and RJ-1) and RPE (RPE-5, RPE-4, RPE-3, RPE-2, and RPE-1) and S-RPE (S-RPE-5, S-RPE-4, S-RPE-3, S-RPE-2, and S-RPE-1). Data showed a large negative correlation between RJ-3 and RPE-4 (r = −0.58. *p* = 0.04), and RJ-1 and RPE-4 (r = −0.62. *p* = 0.03) (See Table 2). A new large negative correlation was found between RJ-5 and S-RPE-4 (r = −0.59. *p* = 0.04), RJ-4 and S-RPE-4 (r = −0.60. *p* = 0.04), RJ-3 and S-RPE-4 (r = −0.64. *p* = 0.02), and RJ-1 and S-RPE-4 (r = −0.60. *p* = 0.03).

### 3.5. Number of Jumps

A new correlation analysis was performed between NJ (NJ-5, NJ-4, NJ-3, NJ-2, and NJ-1) and RPE (RPE-5, RPE-4, RPE-3, RPE-2, and RPE-1) and S-RPE (S-RPE-5, S-RPE-4, S-RPE-3, S-RPE-2, and S-RPE-1), which did not show any correlations.

### 3.6. Density

The last correlation analysis was performed between D (D-5, D-4, D-3, D-2, and D-1) and RPE (RPE-5, RPE-4, RPE-3, RPE-2, and RPE-1) and S-RPE (S-RPE-5, S-RPE-4, S-RPE-3, S-RPE-2, and S-RPE-1), which did not show any correlations.

Posteriorly, a multilinear regression analysis was performed to verify which values of internal intensity could be used to better explain the performance of external intensity variables. Thus, we found our multiple regression for RPE-4 revealed significant effects for MXJ-4, MXJ-3, MXJ-1, RJ-3, and RJ-1 (r^2^ = 0.34, r^2^ = 0.40. r^2^ = 0.41, r^2^ = 33 and r^2^ = 38), respectively. On the other hand, multiple regression for S_RPE-4 showed significant effects for MXJ-5, MXJ-4, MXJ-3, MXJ-1, RJ-5, RJ-4, RJ-3, and RJ-1 (r^2^ = 0.35, r^2^ = 0.39, r^2^ = 0.44, r^2^ = 34, r^2^ = 0.34, r^2^ = 0.35, r^2^ = 0.40 and r^2^ = 36). For more information, see Table 2.

## 4. Discussion

Few studies have analyzed the relationship between internal and external training intensity in volleyball athletes. The present research aimed to analyze: (1) The difference between the internal and external intensity variables, according to the number of days before the match; (2) the correlations between the internal and external intensity variables; and (3) the prediction of the internal intensity from the external intensity variables. The hypotheses of the present research were as follows: (1) Significant differences in external and internal training intensity within the week will be observed, with higher efforts reported in the middle of the week; and (2) internal intensities reported subjectively (e.g., rate of perceived exertion) will present significant correlations with the number of jumps.

The results showed that there was a difference between the microcycle and the RPE, S-RPE, JHA, and NJ. Such findings partially corroborate the study by Lima et al. [31] who showed that there was a greater number of jumps performed on MD-2, when compared to MD-1 (+34.5%). The authors also verified that there were no differences across the other training sessions in relation to the number of days before the competition, as well as in the jump frequency (1.57 jumps/m) and the average (~52%) and maximum jump height (~80%) between training sessions. In another study, Lima et al. [16] observed that RPE was significantly higher on MD-2 and MD-3 compared to MD-1, although no significant difference in s-RPE was found between training days. When considering jumps, the authors noticed that a greater number of jumps were performed on MD-2 than on MD-1 and that no significant differences were found in the jumps per minute or the average and maximum height of the jumps between the training days.

In addition to the goals established by the coach, the internal and external intensity throughout the competition week should be considered to respect the team’s recovery time for the next game [7,32,33]. Thus, the training load is higher at the beginning of the week, remains high throughout the week, and decreases at the week ending allowing players to present their best performance on competition day [15,33,34]. However, when the weeks have congested games, although there is no difference in physical demand when compared to regular weeks [4], the accumulation of games affects the well-being of players compared to regular weeks, even with training loads before games being similar between regular and congested weeks [35]. Therefore, it is necessary that there is an adequate ratio between training and athlete recovery to achieve the intended adaptation for the games throughout the competition [15,36], considering the importance of the game in the sports season [37].

When analyzing the correlations between internal and external intensity, it was observed that there was a negative correlation between the Maximum Jump and the Range Jump with RPE-4 and S_RPE-4, indicating that the greater the perception of effort at 4 days before the game, the lower the height of the maximum jump performed. In addition, these variables showed predictive power over RPE-4 and S_RPE-4. Such findings show that higher internal intensities are correlated with lower external intensities in sessions further away from the game day, a fact not found on the days closest to the game, and it is possible to suggest that there is an oscillation in the training load during the week, allowing the athlete to recover throughout the sessions so that they are recovered on the game day [38]. Similar findings were found by Mendes et al. [39] analyzing soccer athletes; these authors noticed that athlete fatigue, stress level, and internal intensity were significantly higher in the two training sessions immediately after match day. Nevertheless, previous research has shown that the accumulation of several volleyball matches negatively affects the work capacity and players’ well-being [35,40], requiring a reduction in players’ intensity training closer to game day [35]. However, the literature shows that there is a low correlation between jumping performance and fatigue [13,41,42], suggesting that the correlation found in the present research may be a result of the accumulation of low recovery after the game and the training that took place on MD-5 and MD-4, since the reduction of the training intensity within proximity to the game would allow the necessary and intended adaptation by the technical committee, according to the micro and mesocycle planning [15].

In this context, coaches should consider that athletes’ performance in competitions depends on the control and monitoring of physical, technical, tactical, and psychological capacities, making it important to reduce training intensities over the weekly sessions in the competitive period to improve the athletes’ recovery, and, in some periods, these reductions may be insufficient for athletes’ optimal recovery [18]. In addition, the match difficulty level, location, and the number of days before the games must be taken into account in the distribution of weekly training intensities, since they influence training planning and can impact athletes’ recovery [43,44]. In addition, another aspect that affects athletes’ sports performance is related to the accumulation of physical training load, which directly impacts the dynamics of the workload according to the competition phase [45], psychophysiological stress throughout the season that comes from training sessions, as well as excessive travel, a reduction in rest time, and changes in training logistics [18]. In this context, it is not uncommon for the coach to overestimate the game effort and underestimate athletes’ degree of recovery, and this incompatibility can generate inadequate planning of training sessions, leading to reduced performance during games [46].

Despite the valuable results, this study had some limitations: (i) Only one male team was made a part of the study; (ii) the analysis was conducted according to the results of the entire team and not according to the player role; (iii) the small sample size and the number of tests performed to analyze the workload; (iv) the microcycles with a single match and with congested fixtures should be analyzed separately, and (v) fitness training was not expressed in the Borg scale applied to the athletes. Further studies should emphasize the differences between male and female athletes and/or analyze more than one team with similar proficiency.

## 5. Conclusions

The results of this study showed differences between the microcycle and the RPE, S-RPE, JHA, and NJ according to the MD. Respecting the principles of the training fluctuation and supercompensation in single-match or congested-fixture microcycles, training loads that integrate the intensity and extent of training cannot be planned, programmed, and implemented in any other way. Despite the small number of respondents in this study, and according to different contexts, such results could be an important tool for coaches to reflect, plan, monitor, and execute the training unit according to the temporal distance to the competition. Furthermore, the range of instruments available for subjective or objective monitoring is, nowadays, an asset to any coach. Thus, despite the results being limited to a single team, workload monitoring solutions are presented throughout a season.

## Figures and Tables

**Table 1 ijerph-19-08691-t001:** Within-week variations (MD-5, MD-4, MD-3, MD-2, and MD-1) of (i) internal intensity: RPE and S-RPE, and (ii) external intensity: JHA, MJ, MXJ, RJ, NJ, and D (mean ± SD).

							Post-Hoc Comparison									
	MD-5	MD-4	MD-3	MD-2	MD-1	ANOVA	MD-5 vs. MD-4	MD-5 vs. MD-3	MD-5 vs. MD-2	MD-5 vs. MD-1	MD-4 vs. MD-3	MD-4 vs. MD-2	MD-4 vs. MD-1	MD-3 vs. MD-2	MD-3 vs. MD-1	MD-2 vs. MD-1
Rate of Perceived exertion (U.A)	7.47 ± 0.87	6.95 ± 0.76	6.85 ± 0.60	6.78 ± 0.66	6.33 ± 0.58	*p* = 0.01 * η^2^ = 0.36	*p* = 0.05 * d = 0.77	*p* = 0.001 ** d = 1.06	*p* = 0.01 * d = 1.12	*p* = 0.001 ** d = 1.96	*p* = 0.53 d = 0.22	*p* = 0.23 d = 0.33	*p* = 0.001 ** d = 1.33	*p* = 0.56 d = 0.17	*p* = 0.001 ** d = 1.51	*p* = 0.001 ** d = 1.17
Session RPE (U.A)	730.73 ± 110.57	593.26 ± 59.94	592.52 ± 65.15	550.89 ± 62.11	487.73 ± 44.18	*p* = 0.001 ** η^2^ = 0.72	*p* = 0.05 * d = 0.78	*p* = 0.001 ** d = 1.06	*p* = 0.01 * d = 1.12	*p* = 0.001 ** d = 1.96	*p* = 0.53 d = 0.22	*p* = 0.23 d = 0.33	*p* = 0.001 ** d = 1.33	*p* = 0.56 d = 0.17	*p* = 0.001 ** d = 1.51	*p* = 0.001 ** d = 1.17
Jump Heigh Average (cm)	48.81 ± 7.11	50.76 ± 7.72	51.83 ± 7.92	52.31 ± 8.62	52.60 ± 7.69	*p* = 0.02 * η^2^ = 0.30	*p* = 0.13 d = −0.04	*p* = 0.05 * d = −0.05	*p* = 0.03 * d = −0.06	*p* = 0.001 ** d = −0.07	*p* = 0.03 * d = −0.02	*p* = 0.13 d = −0.02	*p* = 0.03 * d = −0.03	*p* = 0.56 d = −0.01	*p* = 0.35 d = −0.01	*p* = 0.72 d = −0.00
Minimum jump (cm)	16.27 ± 2.11	16.33 ± 1.84	16.69 ± 2.25	17.20 ± 3.37	16.90 ± 2.26	*p* = 0.24 η^2^ = 0.12	*p* = 0.78 d = 0.00	*p* = 0.13 d = −0.01	*p* = 0.82 d = 0.00	*p* = 0.82 d = 0.00	*p* = 0.25 d = −0.01	*p* = 0.88 d = 0.00	*p* = 0.96 d = 0.00	*p* = 0.22 d = 0.01	*p* = 0.23 d = 0.01	*p* = 0.97 d = 0.00
Maximal jump (cm)	84.99 ± 15.21	85.38 ± 12.27	86.71 ± 13.11	85.28 ± 11.71	85.33 ± 12.45	*p* = 0.55 η^2^ = 0.06	*p* = 0.001 ** d = 0.02	*p* = 0.001 ** d = 0.02	*p* = 0.001 ** d = 0.02	*p* = 0.001 ** d = 0.03	*p* = 0.95 d = 0.00	*p* = 0.02 * d = 0.01	*p* = 0.001 ** d = 0.04	*p* = 0.05 * d = 0.01	*p* = 0.001 ** d = 0.03	*p* = 0.001 ** d = 0.02
Range jump(cm)	68.72 ± 15.59	69.05 ± 12.53	70.02 ± 13.39	68.08 ± 12.53	68.43 ± 12.62	*p* = 0.54 η^2^ = 0.06	*p* = 0.81 d = 0.00	*p* = 0.18 d = −0.01	*p* = 0.64 d = 0.00	*p* = 0.85 d = 0.00	*p* = 0.44 d = −0.01	*p* = 0.39 d = 0.01	*p* = 0.56 d = 0.00	*p* = 0.13 d = 0.01	*p* = 0.19 d = 0.01	*p* = 0.80 d = 0.00
Number of jumps (n)	104.12 ± 32.67	102.08 ± 31.02	98.55 ± 29.28	93.57 ± 27.20	83.36 ± 18.55	*p* = 0.02 * η^2^ = 0.30	*p* = 0.81 d = 0.00	*p* = 0.53 d = 0.01	*p* = 0.14 d = 0.01	*p* = 0.001 ** d = 0.03	*p* = 0.37 d = 0.00	*p* = 0.03 * d = 0.01	*p* = 0.001 ** d = 0.03	*p* = 0.15 d = 0.01	*p* = 0.001 ** d = 0.02	*p* = 0.001 ** d = 0.02
Density (n of jumps/time of training session)	1.06 ± 0.32	1.20 ± 0.37	1.18 ± 0.34	1.18 ± 0.34	1.10 ± 0.24	*p* = 0.09 η^2^ = 0.19	*p* = 0.07 d = −1.13	*p* = 0.12 d = −1.09	*p* = 0.08 d = −1.11	*p* = 0.47 d = −0.48	*p* = 0.67 d = 0.16	*p* = 0.63 d = 0.14	*p* = 0.12 d = 0.91	*p* = 0.92 d = −0.02	*p* = 0.12 d = 0.85	*p* = 0.08 d = 0.88

* Denotes significance at *p* < 0.05 and ** denotes significance at *p* < 0.01.

**Table 2 ijerph-19-08691-t002:** Values of regression analysis explaining the relevance of different internal intensity variables (RPE and S-RPE).

		R	R^2^	Adjusted R^2^	F	P	SE
Rate of Perceived Exertion-4	Maximal Jump-4	0.58	0.34	0.27	5.24	0.04 *	0.25
	Maximal Jump-3	0.63	0.40	0.34	6.79	0.02 *	0.24
	Maximal Jump-1	0.64	0.41	0.36	7.22	0.02 *	0.24
	Range Jump-3	0.58	0.33	0.27	5.11	0.04 *	0.25
	Range Jump-1	0.62	0.38	0.32	6.26	0.03 *	0.24
Session RPE-4	Maximal Jump-5	0.59	0.35	0.28	5.46	0.04 *	0.25
	Maximal Jump-4	0.62	0.39	0.33	6.42	0.03 *	0.24
	Maximal Jump-3	0.66	0.44	0.38	7.96	0.02 *	0.18
	Maximal Jump-1	0.58	0.34	0.28	5.29	0.04 *	0.26
	Range Jump-5	0.59	0.34	0.28	5.33	0.04 *	0.25
	Range Jump-4	0.59	0.35	0.29	5.56	0.04 *	0.25
	Range Jump-3	0.63	0.40	0.34	6.91	0.02 *	0.24
	Range Jump-1	0.60	0.36	0.30	5.71	0.04 *	0.25

* Denotes significance at *p* < 0.05.

## Data Availability

Not applicable.

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
