# Peer review of "Within-Week Variations and Relationships between Internal and External Intensities Occurring in Male Professional Volleyball Training Sessions"

_ijerph, 2022, doi:10.3390/ijerph19148691_

Round 1

Reviewer 1 Report

In the description of research tools, the used Borg CR10 scale should be described in more detail. 

Were the studies conducted on both workouts during the day - morning and evening? This needs to be clarified.

The conclusions should be more focused on the case study, because with such a small number of respondents, they cannot be applied to the population of volleyball players.

Author Response

In the description of research tools, the used Borg CR10 scale should be described in more detail. 

Were the studies conducted on both workouts during the day - morning and evening? This needs to be clarified.

The conclusions should be more focused on the case study, because with such a small number of respondents, they cannot be applied to the population of volleyball players.

Dear Reviewer, Thank you! We added more detailed information about the Borg CR10.

We also changed the conclusion according to your suggestion.

Reviewer 2 Report

Dear Corresponding Author,

thanks for submitting your paper. I appreciated a lot your work but I have to underline some tricky points. Please read carefully and answer point by point.

1) The abstract is totally unreadble in the part of results. You speal about the RPE-4 and  the S_RPE-4 but it is no description of this parameter so it is hard to comprehend what you are describing. Please, adjust the abstract

2) In the paragraph 2.1 you describe the "match day with training at the same day (MDwT), 104 and the next day of the match (MD+1)" but there is no more trace of these in the rest of the paper. Why? 

3) About the use of the ANOVA in table 1, it could be better to use a post-hoc comparison to underline in which MD (5 to 1) the statistical difference is. Please provide.

4) All the significative correlations are related to the RPE of MD-4, therefore it could be interesting to describe in the paper the content of the tipycal MD-4 training session.

5) in the discussion part it could be better to explain in the details some correlation, such as: how it is justifiable a correlation between RPE-4 and MXJ-1 or RJ-1? It is normal to read correlation between the session MD-4 and the previous (MD-3) and the successive (MD-5) but I don't understand the link with the MD-1. Could you provide more details?

Regards

Author Response

  • The abstract is totally unreadble in the part of results. You speal about the RPE-4 and  the S_RPE-4 but it is no description of this parameter so it is hard to comprehend what you are describing. Please, adjust the abstract.

Dear Reviewer, thank you.

We added the description of the RPE and S_RPE-4 in the abstract.

  • In the paragraph 2.1 you describe the "match day with training at the same day (MDwT), 104 and the next day of the match (MD+1)" but there is no more trace of these in the rest of the paper. Why? 

Dear Reviewer, thank you!

You are completely wright. We opted to exclude that information in the paper.

  • About the use of the ANOVA in table 1, it could be better to use a post-hoc comparison to underline in which MD (5 to 1) the statistical difference is. Please provide.

Thank you very much, the ANOVA was conserved, and we added and controlled the post-hoc comparison

  • All the significative correlations are related to the RPE of MD-4, therefore it could be interesting to describe in the paper the content of the tipycal MD-4 training session.

Dear Reviewer, thank you.

We had no access to the training sessions content, but we reflect that results in the discussion chapter: “When analyzing the correlations between internal and external intensity, it was observed that there was a negative correlation between the Maximum Jump and the Range Jump with the RPE-4 and S_RPE-4, indicating that the greater the perception of effort at 4 days of the game, the lower the height of the maximum jump performed.”

5) in the discussion part it could be better to explain in the details some correlation, such as: how it is justifiable a correlation between RPE-4 and MXJ-1 or RJ-1? It is normal to read correlation between the session MD-4 and the previous (MD-3) and the successive (MD-5) but I don't understand the link with the MD-1. Could you provide more details?

Dear Reviewer, thank you.

We opted to correlate different variables within the microcycle to understand the workload variation during the week. It’s known that the MD-1 is the tapering training session. Thus, we have tried to analyze how the internal and external load varied along the weeks within the season. That’s why our purpose included the analysis between the internal and external intensity variables, according to the number of days before the match; the correlations between the internal and external intensity variables; and the prediction of the internal intensity from the external intensity variables.

Reviewer 3 Report

Since only those microcycles that ended with a match (26 games/26 microcycles) were selected, it is necessary to better clarify the relationship between the number of games and the number of trainings (101 trainings - 3.88 trainings per microcycle).

Was there a multi-game microcycle (2 games or tournament) and how were the days of the week treated in that case?

Considering that a larger number of trainings was obviously realized in some microcycles (the authors state up to 7 trainings), how were situations treated where there was obviously more than one training session per day?

It is necessary to better describe this variable (s-RPE) - from the description of the variable it can be concluded that it corresponds more to the description of load volume (product of subjective feeling of intensity * training duration - definition of load extension) than load intensity.

The authors do not state what happens to the specialized fitness training trainings that are conducted 2 to 4 times a week in the competition period of senior volleyball teams. Fitness training can greatly affect the results expressed on the Borg scale. This is a serious problem because the results obtained do not depend only on the volume of the load on specific volleyball trainings.

The conclusion is too general. According to the rules of the profession,
respecting the principles of the “training wave” and supercompensation
in single-match microcycles, training loads that integrate the intensity
and extent of training cannot be planned, programmed and implemented
in any other way.

Author Response

Since only those microcycles that ended with a match (26 games/26 microcycles) were selected, it is necessary to better clarify the relationship between the number of games and the number of trainings (101 trainings - 3.88 trainings per microcycle).

Dear Reviewer, thank you.

We add additional information to clarify your doubts.

Was there a multi-game microcycle (2 games or tournament) and how were the days of the week treated in that case?

Dear Reviewer, thank you.

In fact, that could be a limitation of our study. We analyzed the data in a general way. Thus, we opted to add that limitation in the final of the discussion chapter.

Considering that a larger number of trainings was obviously realized in some microcycles (the authors state up to 7 trainings), how were situations treated where there was obviously more than one training session per day?

Dear Reviewer, thank you.

In fact, during the microcycle that included more than 5 training sessions, we carefully analyze the data. The morning session was a technical session without jump actions (Service (above the box), reception and set). The only player that had jump actions was the setter, but has the literature says, the jump effort of the setter is much lower than the other playing role, not having influence in their physical fitness between training sessions.

It is necessary to better describe this variable (s-RPE) - from the description of the variable it can be concluded that it corresponds more to the description of load volume (product of subjective feeling of intensity * training duration - definition of load extension) than load intensity.

Dear Reviewer, thank you.

The S-RPE (session RPE) is also considered as training load or load volume, which is the product of the subjective feeling with the training duration). We changed the sentence to clarify the information.

The authors do not state what happens to the specialized fitness training trainings that are conducted 2 to 4 times a week in the competition period of senior volleyball teams. Fitness training can greatly affect the results expressed on the Borg scale. This is a serious problem because the results obtained do not depend only on the volume of the load on specific volleyball trainings.

Dear Reviewer, thank you.

You’re completely wright. Despite that, we didn’t assess the fitness training because we didn’t collect the data regarding to that. However, our purpose was to analyze the volleyball training sessions. In further studies we’ll take in account that problem.

The conclusion is too general. According to the rules of the profession,
respecting the principles of the “training wave” and supercompensation
in single-match microcycles, training loads that integrate the intensity
and extent of training cannot be planned, programmed and implemented
in any other way.

Dear Reviewer, thank you.

We adjust you suggestion to our conclusion. Thank you.

Reviewer 4 Report

The aim of this study was to test the within-week variation of internal and external training intensity outcomes us. The authors should be commended on the obvious level of hard work that was required to collect the amount of data collected in this study. The entire manuscript needs to be proof read as there are issues with the English grammar and sentence structure throughout. I was not specific about these issues in my comments as they are so prevalent. I have additional comments that I believe will improve the manuscript and enhance the readability and interpretation of the results. Please see my specific comments below.

Abstract

L24: This should be 12.

L35: give the magnitude of the correlation.

Introduction

Line 70-73: Specify how many more jumps.

Methods

L92: “To respond to our hypotheses,” can be removed as it is unnecessary.

L93: 14 players? Shouldn’t this be 12.

L95: Write the number as 89,801

L96: Write the number as 11,214

L115: “only players with terminal actions tasks were admitted.” What does this mean?

L131: Please include the relevant reliability and validity statistics from your included citations.

L139: “the density of the training sessions” this needs to be explained further.

L143: An observational and longitudinal study was adopted.” This should be in the Experimental approach to the problem section.

L155-156: Why was the Kolmogorov-Smirnov test used rather than the Shapiro-Wilk’s test?

L164-165: Provide a citation for the thresholds for interpretation of the correlations.

L158: Why not include MD and MD+1?

L170: This should be “the alpha level was set at p<0.05.”

State the sensitivity of the study based in your sample size.

Results

Table 1

Is it convention to report RPEs to two decimal places? Please check the correct number of decimal places for all variables.

Include the variable units in the left most column.

Please check if you have enough space to include the actual names of variables on the left most column and top row rather than acronyms? This would make the table easier for readers to understand.

L180-184: Did you run post-hoc tests to determine where the differences existed?

Tables 3.1 – 3.6: Colour maps are not really appropriate for correlations. You used null hypothesis significance testing so really the response is binary, statistically significant or not, less than 0.05 or greater than .05. You should only highlight the correlations where p<0.05.

Figure 1: The top right plot does not give the r and p value.

Figure 2: Revise the top right plot’s x-axis.

Overall comment: You have performed a very large number of statistical tests and I don’t believe you controlled for multiple comparisons. Please justify this decision.

Discussion: You need to consider the possibility of type 1 errors due to the number of tests performed and the possibility of type 2 errors due to small sample size. The discussion and limitations should be revised to reflect these considerations.

Author Response

Reviewer 4

The aim of this study was to test the within-week variation of internal and external training intensity outcomes us. The authors should be commended on the obvious level of hard work that was required to collect the amount of data collected in this study. The entire manuscript needs to be proof read as there are issues with the English grammar and sentence structure throughout. I was not specific about these issues in my comments as they are so prevalent. I have additional comments that I believe will improve the manuscript and enhance the readability and interpretation of the results. Please see my specific comments below.

Abstract

L24: This should be 12.

Dear Reviewer. Thank you. We analyzed the entire team, which was constituted by 14 players.

L35: give the magnitude of the correlation.

Dear Reviewer, thank you.

We added the information according your suggestion.

Introduction

Line 70-73: Specify how many more jumps.

 Dear Reviewer, thank you.

Information was added according to your suggestion.

Methods

L92: “To respond to our hypotheses,” can be removed as it is unnecessary.

Dear Reviewer, thank you.

Changes have been made.

L93: 14 players? Shouldn’t this be 12.

Dear Reviewer. Thank you. We analyzed the entire team, which was constituted by 14 players.

L95: Write the number as 89,801

L96: Write the number as 11,214

Dear Reviewer, Thank you!

Changes were made.

L115: “only players with terminal actions tasks were admitted.” What does this mean?

Dear Reviewer, thank you.

We have removed that part.

L131: Please include the relevant reliability and validity statistics from your included citations.

Dear Reviewer, thank you.

Information was added to the manuscript.

L139: “the density of the training sessions” this needs to be explained further. “The density of the training session is considered as the frequency of jumps divided for a period of time (number of jumps/time of training session).

Dear Reviewer, thank you.

We explain the density variable in the next sentence.

L143: An observational and longitudinal study was adopted.” This should be in the Experimental approach to the problem section.

Dear Reviewer, thank you!

We have moved the sentence to the Experimental approach.

L155-156: Why was the Kolmogorov-Smirnov test used rather than the Shapiro-Wilk’s test?

Thank you very much. The Shapiro–Wilk’s test was conducted to verify if all data were normally distributed. The analysis section has been updated correspondingly.

L164-165: Provide a citation for the thresholds for interpretation of the correlations.

Dear Reviewer, thank you!

Citation was added to the manuscript.

L158: Why not include MD and MD+1?

Dear Reviewer, thank you. We opted to remove de MD and MD+1 in the manuscript, because it was not the aim of our study.

L170: This should be “the alpha level was set at p<0.05.”

State the sensitivity of the study based in your sample size.

 Dear Reviewer, thank you.

Changes were made according your suggestions

Results

Table 1

Is it convention to report RPEs to two decimal places? Please check the correct number of decimal places for all variables. Done. All the measures were updated with tow decimal places.

Include the variable units in the left most column.Done

Please check if you have enough space to include the actual names of variables on the left most column and top row rather than acronyms? This would make the table easier for readers to understand.

L180-184: Did you run post-hoc tests to determine where the differences existed?

Thank you very much. We added and controlled the post-hoc comparison in results section.

Tables 3.1 – 3.6: Colour maps are not really appropriate for correlations. You used null hypothesis significance testing so really the response is binary, statistically significant or not, less than 0.05 or greater than .05. You should only highlight the correlations where p<0.05.

The correlation tables were deleted, and we complete the text correspondingly.

Figure 1: The top right plot does not give the r and p value.

We add the information in text.

Figure 2: Revise the top right plot’s x-axis.

We add the information in text.

Overall comment: You have performed a very large number of statistical tests and I don’t believe you controlled for multiple comparisons. Please justify this decision.

Thanks you very much for your recomendations. The comments are highly appreciated. In fact, we believe vehemently that the present research is a significant contribution to the existing literature. In this sense, we add the multiple comparison.

Discussion: You need to consider the possibility of type 1 errors due to the number of tests performed and the possibility of type 2 errors due to small sample size. The discussion and limitations should be revised to reflect these considerations.

Dear Reviewer, thank you.

We revised the limitations, and according to your suggestions we included the sample size and the number of performed tests as two more limitations.

Round 2

Reviewer 2 Report

I read the changes. Thanks, now it is clearer

Author Response

Dear Reviewer. Thank you very much for you contribution to improve our paper.

Reviewer 3 Report

1.                  The authors do not state what happens to the specialized fitness training that are conducted 2 to 4 times a week in the competition period of senior volleyball teams. Fitness training can greatly affect the results expressed on the Borg scale. This is a serious problem because the results obtained do not depend only on the volume of the load on specific volleyball trainings.

Answer: Dear Reviewer, thank you. You’re completely wright. Despite that, we didn’t assess the fitness training because we didn’t collect the data regarding to that. However, our purpose was to analyze the volleyball training sessions. In further studies we’ll take in account that problem.

Suggestion: Please, explain the situation with the fitness trainings (and their possible influence on results expressed on the Borg scale) the to the readers and list this as a notable limitation of the study.

2.                  The conclusion is too general. According to the rules of the profession,  respecting the principles of the “training wave” and supercompensation  in single-match micro cycles, training loads that integrate the intensity  and extent of training cannot be planned, programmed, and implemented  in any other way.

Answer: Dear Reviewer, thank you. We adjust your suggestion to our conclusion. Thank you.

Suggestion: You miss word other in the text…Please add.

Author Response

Dear Reviewer, Thank you.

Suggestion: Please, explain the situation with the fitness trainings (and their possible influence on results expressed on the Borg scale) the to the readers and list this as a notable limitation of the study.

 According to your suggestion, we add the sentence: "In addition, another aspect that affects athletes' sports performance is related to the accumulation of physical training load, which directly impacts the dynamics of the workload according the competition phase " exported from the study of :

Thiago A.G. H, Maurício G. Bara F, Bernardo M, Daniel G.S. de F, Jeferson M. V. Season Impact on the Technical and Physical Training Load in Professional Volleyball. Int J Sport Exerc Med 2021;7:1–8.

Additionaly, we added this limitation in our work.

The conclusion is too general. According to the rules of the profession,  respecting the principles of the “training wave” and supercompensation  in single-match micro cycles, training loads that integrate the intensity  and extent of training cannot be planned, programmed, and implemented  in any other way.

Dear Reviewer, thank you.

We added the missing word.

Reviewer 4 Report

The majority of my comments from the previous round of reviews have been addressed, however, several comments have not been addressed adequately.

“Two setters, four middle blockers, four outside hitters, and two opposites” – this adds up to 12 players.

The entire manuscript still needs to be thoroughly revised for English grammar and sentence structure throughout.

Replace r2 with “r2”.

L141-145: The relevant reliability and validity statistics have not been added.

The sensitivity of the study has not been added.

Author Response

Two setters, four middle blockers, four outside hitters, and two opposites” – this adds up to 12 players.

Dear Reviewer. Thank you. Actually, in this study participated 14 players. We have corrected this information.

The entire manuscript still needs to be thoroughly revised for English grammar and sentence structure throughout.

Dear Reviewer, thank you. We'll send the manuscript to proof reading.

Replace r2 with “r2”.

Dear reviewer, thank you. Changes were made.

L141-145: The relevant reliability and validity statistics have not been added.

The sensitivity of the study has not been added.

Dear Reviewer, thank you.

We included 2 more references with the ICC values of validity and reliability of the Borg scale.